# Bayesian hierarchical modeling of joint spatiotemporal risk patterns for Human Immunodeficiency Virus (HIV) and Tuberculosis (TB) in Kenya

Verrah A. Otiende[1]*, Thomas N. Achia[2], Henry G. Mwambi[2]

**1** Department of Mathematical Sciences, Pan African University Institute of Basic Sciences Technology and Innovation, Nairobi, Kenya, **2** School of Mathematics, Statistics & Computer Science, University of KwaZulu Natal, Pietermaritzburg, South Africa

* otiende.verrah@students.jkuat.ac.ke, verrahodhiambo@gmail.com

## Abstract

The simultaneous spatiotemporal modeling of multiple related diseases strengthens inferences by borrowing information between related diseases. Numerous research contributions to spatiotemporal modeling approaches exhibit their strengths differently with increasing complexity. However, contributions that combine spatiotemporal approaches to modeling of multiple diseases simultaneously are not so common. We present a full Bayesian hierarchical spatio-temporal approach to the joint modeling of Human Immunodeficiency Virus and Tuberculosis incidences in Kenya. Using case notification data for the period 2012–2017, we estimated the model parameters and determined the joint spatial patterns and temporal variations. Our model included specific and shared spatial and temporal effects. The specific random effects allowed for departures from the shared patterns for the different diseases. The space-time interaction term characterized the underlying spatial patterns with every temporal fluctuation. We assumed the shared random effects to be the structured effects and the disease-specific random effects to be unstructured effects. We detected the spatial similarity in the distribution of Tuberculosis and Human Immunodeficiency Virus in approximately 29 counties around the western, central and southern regions of Kenya. The distribution of the shared relative risks had minimal difference with the Human Immunodeficiency Virus disease-specific relative risk whereas that of Tuberculosis presented many more counties as high-risk areas. The flexibility and informative outputs of Bayesian Hierarchical Models enabled us to identify the similarities and differences in the distribution of the relative risks associated with each disease. Estimating the Human Immunodeficiency Virus and Tuberculosis shared relative risks provide additional insights towards collaborative monitoring of the diseases and control efforts.

**Data Availability Statement:** The data underlying the results presented in the study are available from NASCOP (P.O. Box 19361-00202, Nairobi-Kenya; telephone: +254-775597297; https://dwh.

nascop.org/) and NLTP (P.O. Box 20781-00202, Nairobi-Kenya; telephone: +254-773977440; http://pms.dltld.or.ke/). Interested researchers may contact NASCOP (info@nascop.or.ke) for the HIV data and NLTP (info@nltp.co.ke) for the TB data. The county population estimates for the period 2012-2017 are available from the Keya National Bureau of Statistics (P.O. Box 30266-00100, Nairobi-Kenya; telephone: +254-20-317583; www.knbs.or.ke).

**Funding:** The authors would like to thank the African Union and the Pan African University Institute of Basic Sciences Technology and Innovation (PAUISTI) for funding the study through VO's Ph.D. work. The funders had no role in study design, data collection and analysis, decision to publish, or preparation of the manuscript.

**Competing interests:** The authors have declared that no competing interests exist

## Introduction

Disease mapping as a modeling approach is commonly used to describe the geographical distribution of disease burden thereby generating hypotheses on their possible causes and differences [1]. Statistical methods for characterization of the geographical variations have contributed significantly towards advancing the focus of disease mapping to incorporate other analysis techniques. An extension to the univariate disease mapping considering a single disease is the joint spatial modeling of multiple related diseases with common risk factors. This extension outspreads the standard univariate disease mapping methodologies by generating both the shared and divergent trends thereby increasing the estimation precision of disease risk [2]. Including the time dimension further advances the subject of disease mapping to the spatiotemporal modeling of the variation of disease risk. Such analyses enables studying of spatial patterns, temporal variations and spatiotemporal interaction thereby giving deeper insights over purely spatial mapping [3,4]. Another extension is the joint spatiotemporal modeling of multiple diseases. This approach advances the space-time analysis to model multiple diseases simultaneously thereby strengthening inference by borrowing information across neighboring regions and between related diseases or sub-populations with common aetiological factors [5]. The benefits of borrowing information lie in the ability to observe concurrency of patterns and to allow conditioning of one disease on others [6] which is very valuable when accounting for uncertainty due to sparse disease count or underreporting [7]

Studies combining spatiotemporal approaches to modeling multiple diseases simultaneously are not so common despite the development and application of novel computational techniques. Reviews from literature show few contributions of these approaches that exhibit their strengths differently with increasing complexities. [8] used a Bayesian factor analysis approach to combine the space-time disease mapping and joint modeling of different cancers. [9] used a full Bayesian hierarchical model to split the disease risks into shared and disease-specific spatiotemporal components. Their definition of multiple diseases was male and female subpopulations. [10] also applied a full Bayesian hierarchical approach to their spatiotemporal model to estimate the relative risk of various cancers while adjusting for age and gender. Using the hierarchical Bayesian factor models, [11] combined the dynamic factor analytic models with space-time disease mapping and produced a flexible framework for jointly analyzing multiple related diseases. Another study by [12] compared three formulations of the spatiotemporal shared component model on five diseases and examined the changes in the shared factors over time.

Incorporating the spatiotemporal modeling approaches to modeling of multiple diseases simultaneously adds significant complexity to the model structure [13]. The Bayesian hierarchical modeling approach makes an appropriate framework for solving complexities in the spatiotemporal structures [9,10,14,15]. The uniqueness of the Bayesian approach that differentiates it from other classical approaches is its robustness to interpretation of the posterior estimates and generating inferences of all model parameters [16].

The feasibility of using case notifications as a surrogate of population-based studies to estimate the shared and disease-specific risks of multiple diseases is also unknown. Against this background, we investigate the spatial and temporal patterns of Human Immunodeficiency Virus (HIV) and Tuberculosis (TB) burden in Kenya jointly using case notification data for a six-year period and characterize the areas with unusually high relative risks. Maps generated from these models describe new exposure hypotheses that warrant further epidemiological investigations on existing challenges and opportunities for disease surveillance and etiology.

### Motivation of the study

The HIV and TB diseases have a co-epidemic overlap in their epidemiologic characteristics and clinical manifestations [17]. Both the HIV and TB pathogens interact synergistically,

accelerating the progress of illness thereby increasing the likelihood of death [18]. TB is a communicable disease and a major cause of ill health globally that affects the lungs (pulmonary) but can also affect other sites (extrapulmonary) [19,20]. Despite being a curable and preventable disease, TB is the leading cause of death from bacterial infection worldwide and undoubtedly representing a global public health priority [21–24]. TB is also the leading cause of morbidity and mortality among people living with HIV/AIDS [24], accounting for approximately 40% of deaths globally [20]. HIV disease is equally a major global health concern causing substantial morbidity, mortality rate, human suffering, and development challenges [25]. HIV infection increases the risk of TB disease up to 20 times [26,27]. The rapid disease progression and associated leading opportunistic infection, TB, contribute to the high mortality rates [28]. Beyond affecting the health of individuals, HIV and TB diseases are stigmatized [17] and have a significant impact on the social, economic and political stability of the hardest-hit countries [29,30].

Kenya suffers from the dual epidemics ranking 4[th] and 15[th] globally in the high disease burden for HIV and TB respectively [31–33]. In 2013, Kenya reported more than 35% of the notified TB to be HIV infected compared to the global 13%. Some regions had up to 75% HIV infections among the TB patients [27]. The high HIV prevalence in Kenya is the major driver of TB related morbidity [33]. The Kenya AIDS Indicator Survey of 2012 indicated that one third of persons who reported prior tuberculosis nationally were HIV positive [34,35]. Equally, TB was the leading opportunistic infection in settings of high HIV prevalence [35,36] and the most re-emerging infection for people living with HIV [37,38]. The KAIS-2012 survey also indicated a stronger association between HIV and TB for the self -reported cases with 1 out of every 5 HIV positive cases having a TB recurrence situation [34].

Studies on TB and HIV incidences in Kenya have limited spatial and temporal scoping making generalizations of their findings to the whole country difficult. This study uses routine case notification data to provide more accurate estimates and insights on the elevated risk areas of HIV and TB individually and jointly over time. By using the spatiotemporal approach to modeling HIV and TB diseases simultaneously, we present the shared and disease-specific spatial risk patterns and explore their temporal evolution. The joint model also determines the combined and disease-specific elevated risk areas.

## Materials and methods

### Study location

The study was conducted in Kenya, a country of great diversity situated in East Africa extending between latitudes $4^0 30N$ and $4^0 30S$ and longitudes $34^0 00E$ and $42^0 00E$ [39]. Kenya has a coastline stretch of approximately 14420 km along the Indian ocean mostly covered by salt-tolerant mangrove trees that creates a distinctive ecological zone [40]. The total coverage area of Kenya is $583,367km^2$ with $569,140km^2$ as land area and $14,227km^2$ being water area [41]. The diversity of Kenya's landscape is shaped by four distinguishable relief zones; these are the coastal and eastern broad plains, the central and western highlands, the Rift Valley and Lake Victoria basins [42].

Kenya is administratively subdivided into 47 counties as the first level of administrative subdivisions which in turn are further subdivided into 290 sub-counties and 1450 wards as the second and third levels of administrative subdivisions respectively [43]. The population of Kenya is unevenly distributed throughout the country and predominantly urban. The population density has continued to increase from 77.9% in 2012 to 88.2% in 2017 per sq.km [44–46]. Supplementary information S2 presents the geospatial arrangements and the list of the 47 counties of Kenya according to their corresponding geographic codes as used in this study. The population estimates per county from 2012–2017 are in supplementary information S3.

A big obstacle to human capital development in Kenya is health challenges. Many people are exposed to a wide range of disease burdens largely because of the country's geographical, economic, political and climatic conditions [47] which are further compounded by inadequate resources to mitigate the impact of health risks [48]. Presently, cases of emerging and re-emerging diseases–like TB and HIV—are on the rise thereby having important implications on public health policy processes [49]. Understanding variations in incidence trends at the county level–where health services are planned, organized and delivered–is essential in addressing health inequalities.

## Data sources

This study considered routine case notification data on TB and HIV diseases in 47 counties of Kenya for 6 years, 2012–2017. Case notifications are data from specific subpopulations who seek treatment and care from health facilities; these are geographically representative of nearby populations. The data were collected and made available by two main sources; the National Tuberculosis Leprosy & Lung Disease Program (NTLD-P) and National AIDS & STIs Control Program (NASCOP). The Government of Kenya has routine case-based monitoring and reporting regulations for TB and HIV diseases through the NLTP and NASCOP programs. The Integrated Electronic Medical Records (EMR) Data Warehouse (IDWH) is the on-line case-based repository hosted by NASCOP accommodating all EMR databases from all health facilities across the country. It operates both as a repository and analytics platform presenting data through interactive dashboards and ad-hoc data analysis. Health facilities update their EMR databases into the IDWH on a monthly basis. The Tuberculosis Information from Basic Unit (TIBU) is the centrally located case-based surveillance system hosted by NLTP that allows for real-time reporting. Since its inception in 2012, TIBU has made notifications of TB patients very timely and instant in report generation. All public, faith-based and private treatment centers in the country enter data into the TIBU system. There are 301 TB control zones across the 47 counties of Kenya which are coordinated by the Sub County TB and Leprosy Coordinators (SCTLCs), who are responsible for notifying TB cases from health facilities in their control zones into the TIBU system. Both programs have adapted the data recording and reporting standards of WHO at the health facilities in every county and the national surveillance system. For the purpose of this study, we analyze the data aggregated per county per year for the period 2012–2017.

## Variables in the study

The study involved three variables; HIV case notifications, TB case notification and Population estimates. Each of this variable was aggregated at county level for each year of study. The HIV and TB case notifications were the total number of diagnosed and reported cases within the national surveillance system. The county yearly population estimates were projected from the 2009 population census.

## Ethical considerations

This study involved the use of non-identifiable secondary data collected as part of the routine programs monitoring. Ethical permission to use the routinely collected data was obtained from NASCOP and NLTP, which are commissioned by the Ministry of Health to host the data surveillance systems for the HIV and TB programmes respectively. The study was subjected to Human Research Protection Review by the African Medical Research Foundation (AMREF Health Africa) Ethical and Scientific Review Committee (ESRC), which determined it not to constitute human participation.

## Hierarchical model specification

We formulated a statistical model that applied the spatiotemporal methods to the modeling of HIV and TB simultaneously. The study applied a full Bayesian Hierarchical approach on the model to estimate the spatial and temporal parameters for the two diseases individually and jointly. We applied the spatial and temporal specifications of [9] and [50] to our model. While [9] modeled a single disease for two subpopulations, we modeled two diseases -HIV and TB- with co-epidemic overlap. They were also keen to interpret the unexpected differential risks between two subpopulations; our aim was to determine the shared and specific spatiotemporal patterns to interpret the relationships between the two diseases. Unlike [50], our model accounts for the joint space-time interaction using similar specifications as [9].

**a. Log-linear model.**   The initial step in defining our model within the Hierarchical Bayesian framework was selecting the probability distribution for the observed data. This study utilized the Poisson distribution from the exponential family. For county s in the year t, we modeled cases notification $y_{dst}$ for disease d, where d = 1 is HIV and d = 2 is TB as;

$$y_{dst} \sim Poisson(\mu_{dst}), \quad \mu_{dst} = \rho_{dst}E_{dst}$$

This study defined the mean $\mu_{dst}$ in terms of the unknown relative risk $\rho_{dst}$ and the expected number of cases $E_{dst}$. We computed $E_{dst}$ per county per year. Our statistical consideration for the standard population $N_s$ was the average of the pooled county population estimates, i.e. $N_s = \frac{P_s}{T}$, where P is the estimated population of county s, which we are considering to be the population at risk of disease d and T is the number of years, which is six years for this study. We calculated the crude rate as $R_{dst} = \frac{\sum y_{dst}}{P_{st}}$, where $\sum y_{dst}$ and $P_{st}$ are the number of cases for disease d and estimated population respectively for county s in the year t. We then multiplied the crude rate by the standard population $N_s$ to obtain the expected number of cases for disease d for county s in the year t as;

$$E_{dst} = R_{dst} \times N_s$$

The linear predictor of the unknown relative risk was on the logarithmic scale, $\eta_{dst} = \log(\rho_{dst})$ which is the recommended invertible link function for the Poisson family of distributions. The variability of the cases around the unknown relative risks $\rho_{dst}$ for HIV and TB respectively were as follows;

$$y_{1st} \sim Pois(\rho_{1st}E_{1st}) \quad \eta_{1st} = \alpha_1 + \lambda_s\delta + \xi_t\kappa + \beta_{1s} + \gamma_{1t} + \upsilon_{st}$$

$$y_{2st} \sim Pois(\rho_{2st}E_{2st}) \quad \eta_{2st} = \alpha_2 + \frac{\lambda_s}{\delta} + \frac{\xi_t}{\kappa} + \beta_{2s} + \gamma_{2t} + \upsilon_{st}$$

The linear predictor was defined by the following terms; the shared spatial effect, ($\lambda = \{\lambda_s\}_{s=1,2,\ldots,S}$), the disease-specific spatial effect ($\beta = \{\beta_{ds}\}_{d=1,2; s=1,2,\ldots,S}$), the shared time trend ($\xi = \{\xi_t\}_{t=1,\ldots,T}$), the disease-specific time trend ($\gamma = \{\gamma_{dt}\}_{d=1,2; t=1,\ldots,T}$), and the space-time interaction term ($\upsilon = \{\upsilon_{st}\}_{s=1,\ldots,S; t=1,\ldots,T}$). The notations $\alpha_d$, $\lambda_s$, and $\xi_t$ captured disease-specific intercept, space and time main effects respectively whereas $\gamma_{dt}$ and $\beta_{ds}$ were disease-time and disease-space interactions of order 2 respectively. The coefficients $\delta$ and $\kappa$ represented the spatial and temporal scaling parameters on the shared term to the risk of TB compared to HIV. Even though the overall relative risk level is the same for both diseases, the magnitude of the area-specific and time-specific relative risks may differ—hence the need for the scaling parameters [9,51,52]. Therefore, contribution of the shared component to the

overall relative risk is weighted by the scaling parameters to allow different risk gradients for each disease.

We applied a symmetric formulation to both the shared and disease-specific random effects, implying that $\lambda_s$, and $\xi_t$ captured the common spatial and temporal patterns. The terms $\gamma_{dt}$ and $\beta_{ds}$ allowed for departures from the shared patterns for the different diseases. The space-time interaction term, $\nu_{st}$ provided additional flexibility towards identifying varying patterns. We assumed the shared random effects to be the structured effects and the disease-specific random effects to be unstructured effects.

**b. Bayesian prior specification.** In a Bayesian framework, random effects are unknown quantities assigned to prior distributions that reflect any prior knowledge on the structure of the effects. The model assigned priors and generated the posterior distribution used for deriving the conditional densities for posterior sampling.

The spatial random effects $\lambda_s$ and $\beta_s$ assumed a spatially correlated prior distribution (CAR spatial priors) with a neighborhood matrix W defined by contiguity; $\tau_\lambda$ and $\tau_\beta$ were the prior hyperparameters. The CAR spatial prior defines a binary specification for the geographical contiguity such that correlation is certain for geographically adjacent areas that is:

$$W_{sj} = \begin{cases} 1 & \text{if } s \sim j \\ 0 & \text{otherwise} \end{cases}$$

The non-contiguous areal units are conditionally independent given the values of the remaining random effects. The $s^{th}$ diagonal elements are the number of neighbors of the $s^{th}$ region, therefore $\sum W_{sk} = 0, \forall s$.

To reflect the prior of yearly fluctuations for $\xi_t$ and $\gamma_{dt}$, the study assumed a random walk prior of order 1 (RW (1)) with a weighted matrix Q which defines the temporal neighborhood with $\tau_\xi$ and $\tau_\gamma$ as the prior hyperparameters. The expression for the random walk of order 1 (RW (1)) given a set of temporal random effects $\Theta = \{\theta_t\}$, where $\theta$ represents $\xi$ and $\gamma$ and t signifies the number of equally spaced time points is:

$$\theta_t | \theta_{(-t)} \quad \sim \quad Normal(\theta_{(t+1)}, \sigma_\theta^2) \quad \text{for } t = 1;$$

$$\sim \quad Normal\left(\frac{\theta_{(t-1)} + \theta_{(t+1)}}{2}, \frac{\sigma_\theta^2}{2}\right) \quad \text{for } t = 2, \ldots, T-1;$$

$$Normal\left(\theta_{(t-1)}, \sigma_\theta^2\right) \quad \text{for } t = T;$$

Where $\theta_{-t}$ denotes all elements of $\Theta$ except the $\theta_t$. This is equivalent to specifying

$$\theta_t \mid \theta_{-t} \quad \sim \quad Normal\left(\sum_k C_{tk}\theta_k, \tau_\theta M_{tt}\right) \quad \text{for } t = 1, \ldots, T,$$

$$\text{where } C_{tk} = \frac{Q_{tk}}{Q_{t+}}; \quad Q_{t+} = \sum_k Q_{tk};$$

$$Q_{tk} = \begin{cases} 1 & \begin{aligned} k &= (t-1) \\ k &= (t+1) \end{aligned} ; \quad M_{tt} = \frac{1}{Q_{t+}}. \\ 0 & \text{otherwise} \end{cases}$$

We impose a sum-to-zero constraints on both the spatial and temporal random effects to minimize any identifiability problem on the intercept [9,53].

The space-time interaction term $\nu_{st}$ prior was a simple exchangeable hierarchical structure $\nu_{st} \sim N(0, \sigma_\nu^2)$, where $\sigma_\nu^2 = \frac{1}{\tau_\nu}$. We defined improper flat priors for the intercept as $P(\alpha_d) \propto 1$. The logarithm of the scaling parameters $\delta$ and $\kappa$ assume normal priors $N(0, \sigma_\delta^2)$ and $N(0, \sigma_\kappa^2)$ which are symmetric around zero on the log-scale, therefore, any value is as equally likely as the reciprocal value and the posterior distribution of the relative risks for each disease are exactly the same for $\delta > 0$ and $\kappa > 0$. More precisely,

$$P(\delta_l \leq \delta \leq \delta_u) = P\left(\frac{1}{\delta_u} \leq \frac{1}{\delta} \leq \frac{1}{\delta_l}\right)$$

$$P(\kappa_l \leq \kappa \leq \kappa_u) = P\left(\frac{1}{\kappa_u} \leq \frac{1}{\kappa} \leq \frac{1}{\kappa_l}\right)$$

For the distribution of the hyper-parameters, we assumed the default specifications of INLA whereby we assigned minimally informative priors on the log of the precision of both the structured and unstructured effects, logGamma(0.5, 0.0005). INLA estimates the posterior marginal distribution for the hyperparameters using an integration-free algorithm described in [54]. We use INLA approach for the model estimations as it is capable of handling complex models with large predictor spaces. Equally the approach does not require convergence checking (unlike McMC) [55,56] as it does not suffer from slow convergence and poor mixing.

## Results

To determine the areas of high risks, we created the spatial maps of the standardized incidence ratio for HIV (Fig 1) and TB (Fig 2). The two diseases displayed varying spatiotemporal patterns, though most of the regions of high risk for HIV were also high risk for TB. The progression of the risk during the period 2012–2017 was much faster in TB as compared to HIV.

We considered the analysis of the combined spatial patterns in the model. Table 1 provides the summary statistics of the shared and disease-specific spatial effects. The disease-specific estimates for $\alpha_1$ and $\alpha_2$ were significantly different from zero (as depicted by the credible intervals). These estimates, on average, were greater than 1indicating that the country is still at a high risk of new infections for both diseases. For the precision parameters, the percentage of the variation expresses what each of the parameters fitted contributes to the explained variability in the model. The results show both the spatial and temporal shared components explain

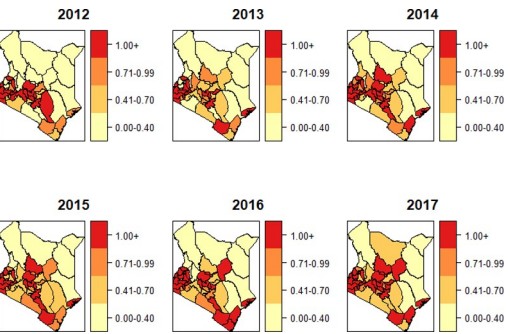

**Fig 1. Standardized incidence ratio $\left(\frac{O_{dst}}{E_{dst}}\right)$ for HIV.**

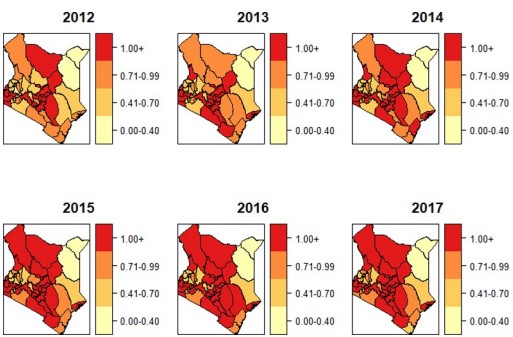

**Fig 2. Standardized incidence ratio $\left(\frac{O_{dst}}{E_{dst}}\right)$ for TB.**

the most variability (comparative variation in the shared parameters relative to the disease-specific parameters).

Fig 3 presents the relative risk of the combined spatial patterns for HIV and TB. The combined spatial patterns are both the structured and unstructured spatial effects. Both HIV and TB had similar areas of high risk in the South West, Central and South regions of Kenya. Further, TB revealed additional areas of high risk in the North West and North of Kenya.

The relative risks for the shared and disease-specific spatial patterns are presented in Fig 4. In all the maps, common areas of high risk were in the west part of the country. Equally, the shared spatial pattern showed fewer regions of high risk compared to TB-specific spatial patterns. This could be because of higher dependence of HIV on the shared spatial term making the shared pattern to account for most of HIV spatial patterns.

The posterior means of the shared, disease-specific, and combined temporal trends are in Fig 5. The shared temporal effect displayed the overall constant increasing risk trend in time with estimates ranging between 0.8–1.1. The HIV temporal trend equally exhibited increasing risk over time with relative risks between 0.8–1.2 whereas the TB temporal trend presented a nearly constant trend across the years with relative risks close to one. The combined temporal trends for HIV and TB showed an increasing risk trend for both diseases. From the combined temporal trends graph, HIV risk was relatively lower than TB in 2012 and 2013. From 2015–2017, the HIV risk trend surpassed that of TB. The shared temporal effect shows trends on the joint diseases that are similar to the disease-specific and combined temporal trend.

**Table 1. Summary statistics of the shared and disease-specific spatial and temporal effects.**

| Parameters | mean(95%CI) | percentage of variation |
|---|---|---|
| **fixed effects:** | | |
| $\alpha_1$ | 1.93(1.86–2.01) | - |
| $\alpha_2$ | 2.42(2.39–2.46) | - |
| **random effects:** | | |
| $\tau_{\beta_{1s}}$ | 0.43(0.24–0.64) | 15% |
| $\tau_{\beta_{2s}}$ | 0.04(0–0.23) | 1% |
| $\tau_{\lambda_s}$ | 0.76(0.58–0.89) | 27% |
| $\tau_{\gamma_{1t}}$ | 0.31(0.01–0.38) | 11% |
| $\tau_{\gamma_{2t}}$ | 0.02(0–0.41) | 1% |
| $\tau_{\xi_t}$ | 0.97(0.88–1) | 34% |
| $\tau_{\upsilon_{st}}$ | 0.31(0.22–0.39) | 11% |

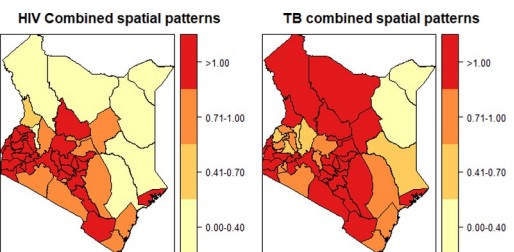

**Fig 3. Relative risk of the combined spatial patterns ($\lambda_s + \beta_{ds}$).**

The posterior probabilities for the smoothed joint Spatio-temporal interaction $P(e^{(v_{st})} > 1|y)$ are in Fig 6, and the posterior estimates are presented in Table 1. Over the years the uncertainty associated with the posterior estimates (exceedance risk) varied for the different counties although most consistent around the coastal region.

## Discussion

We presented a Bayesian Hierarchical approach to the joint modeling of spatio-temporal routinely collected health data. The joint modeling approaches have yielded substantial co-dynamic insights via mathematical, statistical and computational approaches [38]. By optimizing the spatial scale at different points in time, spatial heterogeneity influences the interpretation of temporal patterns more especially in disease dynamics and surveillance [38]. This is especially true for the case of HIV and TB that have significant geographic overlap and are subject to diverse regional variations in their co-dynamics. HIV and TB rank as the leading causes of death from infectious diseases globally with an estimated 2.5 million new HIV infections and 8.7 million incidences of TB annually [57,58]. They have a close link even though their biological co-existence and co-dynamics vary regionally with much burden in Sub-Saharan Africa [33,59]. This study determined the space-time joint risk trends of HIV and TB in Kenya. Our model enabled us to define the shared and specific spatial and temporal patterns of HIV and TB thus identifying similarities and differences in the distribution of the relative risks associated with each disease. The model separately estimated the shared and disease-specific relative risks and displayed the spatial-disease, temporal-disease, and spatio-temporal disease interaction effects across all regions. We included scaling components on the shared spatial and temporal parameters to compare their strength signals for HIV and TB.

The disease-specific spatial and temporal patterns detected areas with varying spatial trends and temporal variations for each disease. The HIV high-risk areas were to the further west of Kenya spreading towards the central and further south. The TB high-risk areas were similar to the HIV high-risk areas but also spread upwards towards the North. The TB geographical progression in relation to HIV was proportionally higher which could reflect environmental factors favoring the TB spread in the high-density settlements especially towards the North. These findings are corroborated in other studies by [33,60]. Looking beyond Kenya, studies by

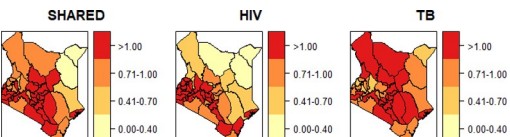

**Fig 4. Relative risk of shared spatial patterns ($\lambda_s$) and disease specific spatial patterns ($\beta_{ds}$).**

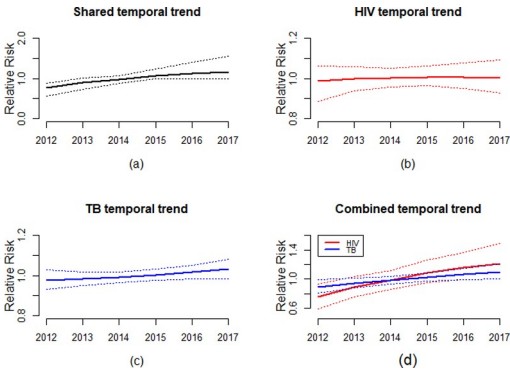

**Fig 5. Relative risk of the shared $\xi_t$, specific $\gamma_{dt}$ and the combined temporal trends ($\xi_t + \gamma_{dt}$).**

[37] revealed that TB appeared to outpace HIV in Rwanda and Burundi while HIV greatly outpaced TB in Mauritania, Senegal and The Gambia.

Joint temporal analysis is important when investigating the temporal coherency of epidemiological trends from the same area [37]. In our study, the shared temporal trend an almost constant risk with minimal variation over time. The disease-specific and combined temporal trends equally presented an increased risk over time. The temporal trend of HIV risk was lower than that of TB for the years 2012 and 2013 but between 2015 and 2017 the HIV risk was higher than TB risk. Similar studies in Sub-Saharan Africa that utilized routinely collected data observed similar temporal dynamics [61–64]. A possible explanation could be HIV drives TB related incidences, therefore, the incidence and prevalence of TB increases (decreases) with increasing (decreasing) HIV trends [65–67].

Our study successfully detected the spatial similarity in the distribution of TB and HIV in approximately 29 counties around the western, central and southern regions of Kenya. The spatial patterns were largely similar for Homabay, Siaya, Kisumu, Busia and Migori counties as the high risk with Mandera, Wajir and Garissa counties at low risk for both HIV and TB. The distribution of the shared relative risks had minimal difference with the HIV disease-specific relative risk whereas that of TB presented many more counties as high-risk areas. This could be attributed to higher dependence of HIV on the shared spatial term making the shared pattern account for most HIV spatial patterns. Similar studies by [68] in China and [60] in Uganda observed significantly persistent clusters for TB and HIV using the spatial co-clustering approach. They examined the clusters exhibited by each disease as well as the combined.

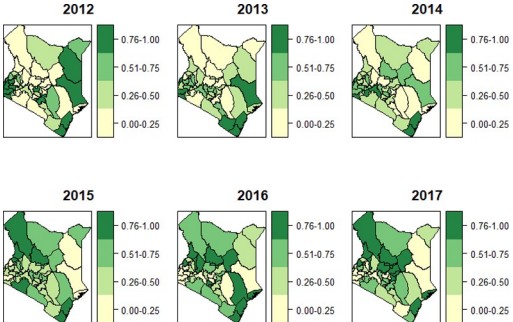

**Fig 6.** Posterior probabilities for the joint spatio-temporal interaction $P(e^{(v_{st})} > 1|y)$.

Our model also presented areas of elevated joint risk by looking at the posterior probabilities of the relative risk to understand the certainty of the shared geographical patterns. There was strong evidence of disease-time, disease-space and space-time interactions. Studies by [68] and [69] also found a strong association on the joint risks of HIV and TB, they used bivariate maps to show that the joint distribution for both TB and HIV diseases was spatially heterogeneous across Brazil.

The primary limitation of the study is using routine case notification data as a surrogate measure of the general population of infected persons; this type of data has challenges of underreporting. The spatial regions were based on the counties since there was no data for sub-county or health facility levels. The age and sex standardized rates were not utilized as the HIV data was not disaggregated by age and gender. Whereas this kind of data is not completely spatially random for the joint epidemic burden, it still captures the spatiotemporal patterns of incidence risk, which is the ultimate goal of this study.

## Conclusions

We defined a joint Bayesian space-time model of two related diseases to jointly quantify the risk of TB relative to HIV thereby facilitating the comparative benefits obtained across populations. The disease burden was apparent at each spatial level of analysis. Identifying the spatial and temporal similarities between HIV and TB enabled us to understand the shared risk. The flexibility and informative outputs of Bayesian Hierarchical Models played a key role in clustering these risk areas. Quantifying how HIV and TB varied together provided additional insights towards collaborative monitoring of the diseases and control efforts. To control the HIV-TB twin epidemic it is important to determine the TB life history that could greatly reduce TB incidences if intervened and which stages of HIV needs optimized TB control benefits.

## Supporting information

**S1 Text. R codes used in the analysis.**
(R)

**S2 Text. List of counties in Kenya.**
(DOCX)

**S3 Text. County population estimates 2012–2017.**
(DOCX)

**S4 Text. TB data.**
(CSV)

**S5 Text. HIV data.**
(CSV)

## Acknowledgments

The authors sincerely acknowledge the National Tuberculosis, Leprosy and Lung Cancer Program (NLTP) and the National AIDS and STI Control Program (NASCOP) for their cordial support in providing the data used in this study.

## Author Contributions

**Conceptualization:** Verrah A. Otiende, Thomas N. Achia, Henry G. Mwambi.

**Data curation:** Verrah A. Otiende, Thomas N. Achia.

**Formal analysis:** Verrah A. Otiende, Thomas N. Achia.

**Methodology:** Verrah A. Otiende, Thomas N. Achia, Henry G. Mwambi.

**Supervision:** Thomas N. Achia, Henry G. Mwambi.

**Validation:** Verrah A. Otiende, Henry G. Mwambi.

**Writing – original draft:** Verrah A. Otiende.

**Writing – review & editing:** Verrah A. Otiende, Thomas N. Achia, Henry G. Mwambi.

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
