## [Decision Letter · Decision Letter 0]

15 Oct 2019

PONE-D-19-25050

Bayesian hierarchical modeling of joint spatiotemporal risk patterns for Human Immunodeficiency Virus (HIV) and Tuberculosis (TB) in Kenya

PLOS ONE

Dear Ms Otiende,

Thank you for submitting your manuscript to PLOS ONE. After careful consideration, we feel that it has merit but does not fully meet PLOS ONE’s publication criteria as it currently stands. Therefore, we invite you to submit a revised version of the manuscript that addresses the points raised during the review process.

Please note that both reviewers have made numerous suggestions to improve the manuscript. In particular, both have noted that the model's construction should be better detailed and discussion of the results should be expanded. Please also note that PLOS journals require authors to make all data underlying the findings described in their manuscript fully available without restriction. The data should be included as part of the manuscript or supplementary information, or should be available in a public repository.

We would appreciate receiving your revised manuscript by Nov 29 2019 11:59PM. To enhance the reproducibility of your results, we recommend that if applicable you deposit your laboratory protocols in protocols.io, where a protocol can be assigned its own identifier (DOI) such that it can be cited independently in the future. For instructions see: http://journals.plos.org/plosone/s/submission-guidelines#loc-laboratory-protocols

We look forward to receiving your revised manuscript.

Kind regards,

Eric Forgoston

Academic Editor

PLOS ONE

Journal Requirements:

2. Please outline in the Methods section how the data was collected, in enough detail for another researcher to reproduce the findings. For instance, please outline the relevant variables for which data was collected. Please also amend your Data availability statement to provide the relevant contact details for where other researchers may apply to access the data. Are you able to provide a list of the counties for which data was collected, in order to ensure reproducibility?

3. Please include your ethics statement in the Methods section, and clarify whether the ethics committee specifically approved the study.

4. We note that Figures 1-6 in your submission contain map/satellite images which may be copyrighted. All PLOS content is published under the Creative Commons Attribution License (CC BY 4.0), which means that the manuscript, images, and Supporting Information files will be freely available online, and any third party is permitted to access, download, copy, distribute, and use these materials in any way, even commercially, with proper attribution. For these reasons, we cannot publish previously copyrighted maps or satellite images created using proprietary data, such as Google software (Google Maps, Street View, and Earth). For more information, see our copyright guidelines: http://journals.plos.org/plosone/s/licenses-and-copyright.

You may seek permission from the original copyright holder of Figures 1-6 to publish the content specifically under the CC BY 4.0 license. 

If you are unable to obtain permission from the original copyright holder to publish these figures under the CC BY 4.0 license or if the copyright holder’s requirements are incompatible with the CC BY 4.0 license, please either i) remove the figure or ii) supply a replacement figure that complies with the CC BY 4.0 license. Please check copyright information on all replacement figures and update the figure caption with source information. If applicable, please specify in the figure caption text when a figure is similar but not identical to the original image and is therefore for illustrative purposes only.The following resources for replacing copyrighted map figures may be helpful:

Reviewers' comments:

Reviewer's Responses to Questions

**Comments to the Author**

1. Is the manuscript technically sound, and do the data support the conclusions?

Reviewer #1: Partly

Reviewer #2: Yes

2. Has the statistical analysis been performed appropriately and rigorously? 

Reviewer #1: No

Reviewer #2: Yes

3. Have the authors made all data underlying the findings in their manuscript fully available?

Reviewer #1: No

Reviewer #2: No

4. Is the manuscript presented in an intelligible fashion and written in standard English?

Reviewer #1: Yes

Reviewer #2: Yes

5. Review Comments to the Author

Reviewer #1: SUMMARY/OVERALL COMMENTS

The paper “Bayesian hierarchical modeling of joint spatiotemporal risk patterns for Human Immunodeficiency Virus (HIV) and Tuberculosis (TB) in Kenya” describes a spatiotemporal model to simultaneously model two diseases that have co-epidemic overlap. While the model itself is not necessarily novel, the application of the model to HIV with TB in Kenya is new. The model needs to be specified more explicitly and laid out better. This application strategy and learning new details about the patterns should be discussed even more than they are. There are some grammatical errors throughout that need to be fixed and I do not report them here. Specific comments are next.

SPECIFIC COMMENTS:

Abstract

1. First sentence. I would not call spatiotemporal modeling of multiple diseases a “recent extension” because it goes back to at least Knorr-Held and Best, 2001.

Introduction

2. I agree that the Bayesian structure gives more interpretable estimates, but what makes it useful for mapping multiple diseases is the ability to leverage information between diseases or sub-populations. This point that the robustness of the Bayesian model makes it useful in mapping a single disease for multiple sub-populations or multiple diseases with common risk factors should be discussed more. Explain why it is important to be able to do this, and the benefits of borrowing strength across diseases or sub-populations.

Motivation

3. Lines 82-85. It would be helpful to have more discussion on similarities between HIV and TB. Jointly modeling both diseases makes strong assumptions about the transmission dynamics and spread of both diseases, and those assumptions are not discussed.

4. Line 87. “HIV disease is equally a major global health cause …”. The word such as “concern” should come between the words health and cause.

5. Lines 90-91. It seems odd to group transmission dynamics and prevention options together. Expand upon current prevention options that might influence transmission dynamics. Can prevention options be incorporated into the statistical model?

Data Sources

6. More details on the counties in Kenya would be helpful for an international audience. Consider mentioning the population size and geographic size (eg., median, range).

7. More detail about which people might be included in the case notification data would be helpful. There is some mention of this in the discussion, but it would make more sense to describe who is represented by case notification data when describing and introducing the data.

Hierarchical Model

8. Line 105. The first sentence is unclear as it mentions combining spatio-temporal concepts. In actuality, it is the joint spatio-temporal modeling of two infectious diseases. What is being combined are the underlying spatial and temporal structures. The authors should be more precise with the terminology being used.

9. Line 108. What does your model borrow from, and how does your model differ from that of [22]? It is stated in the paper that methods are borrowed from [4] and [22] without saying which methods.

Log-linear model

10. Line 117. It should be mentioned that the subscript d stands for disease and to clarify that d=1 is HIV and d=2 is TB or vice-versa.

11. Line 120. It would make more sense for the equations with the linear predictors (lines 130 and 131) to appear after the sentence “The linear predictor … family of distributions” because it is easier to follow the model formulation after seeing the equations.

12. Line 120. How are the expected counts calculated? Knowing this would help in the interpretation of the relative risk.

13. Lines 122-123. When describing the model formulation, instead of writing “specific patterns” or “specific random effects”, it would be clearer to use the phrase “disease-specific patterns” or “disease-specific effects”.

14. Line 126. Talking about random effects is not standard in the Bayesian framework because all parameters have distributions and are thus considered to be “random.”

15. When discussing the model formulation, it would be helpful to give the symbols for each effect specified. For example, “the disease-specific effects, α¬d, β¬ds, and γdt allowed for departures …” Consider moving the first paragraph of the Bayesian model specification right after introducing the model. Furthermore, the spatial correlation and temporal correlation sections introduce parameters that are not part of the model. Because of that, the paper does not flow and it is hard to follow what parameters have what distributions. Consider restructuring to make it easier to follow.

16. It is a very strong assumption to have two diseases share the δ and κ terms where one model includes the reciprocal. More discussion and justification of this formulation is needed.

17. Be specific what is meant by structured and unstructured effects. Presumably the structured effects have the ICAR model and the unstructured effects have independent normal priors, but this should be explicitly stated.

Bayesian Model Specification

18. Line 157 is confusing. Please rephrase.

19. Following from comment 15, it would make more sense to move the part of the Bayesian model specification section discussing the spatial effects to the spatial correlation section and the part discussing the temporal effects to the temporal correlation section.

20. When the ICAR weights in the spatial correlation section are introduced, the S notation is used in the neighborhood matrix. In the Bayesian model specification section the W notation is used. Notation should be consistent.

21. The Bayesian model specification section seems incomplete. There is no discussion of prior distributions for the α parameters or for the hyperparameters (τλ, τβ, τξ, τγ, τν), no discussion of convergence assessment, and no discussion of computation. It looks like from the code that the authors used INLA - this should be mentioned in the manuscript.

22. It is not clear what the hyperparameters for the spatial effects are. Consider adding the ICAR distribution to the spatial correlation section to identify that these hyperparameters come from the variance of the ICAR model.

23. When talking of the hyperparameters of the temporal effects, it is confusing because you used φ in the temporal correlation section, but then switched to τ in the Bayesian model specification section. Notation should be consistent.

Results

24. All the maps need more descriptive labels – it is impossible to know which part of Fig 1 is HIV or TB. And without reading the text you wouldn’t know the maps are on the relative risk scale because they that are labelled as the “effect” not the exponential of the effect.

25. The switch from relative risk to the term standardized incidence ratio seems unnecessary and without explanation. It would be more clear to state that you are plotting exp(η) as the relative risk.

26. The color scale for the maps seems very arbitrary. Relative risk is a ratio – so therefore a 0.71 – 0.99 decrease is comparable to a 1.01 - 1.4 increase. Coloring all relative risks above 1 the same is extremely misleading and uninformative. Relative risks of 1 just mean the risk is what you would expect, it doesn’t make sense for this to be the darkest red.

27. The authors state “the progression of the risk during the period 2012 – 2017 was much faster for TB as compared to HIV” based only on looking at the maps. You can use the posterior distributions of γ1t and γ2t to compute the posterior probability that the progression of risk for TB is faster than the progression of risk for HIV. This would be much more informative.

28. The labels for figure 2 and 3 say posterior means of each effects are plotted, but they are actually plotting those effects on the relative risk scale (exponentiated). The labels should indicate this.

29. Indicate on Figure 4 that plots show exponentiated values by labeling the y-axis. It is confusing what is being plotted as the combined temporal effect, because for both HIV and TB in 2012 the shared effect is above 1 and the combined effect is less than the disease specific effects for both diseases, but the claim is that this is plotting the sum? Looking in the R code it does not appear that the authors plotted the correct thing either for the shared effect or for the sum of the shared and disease specific effects.

30. In Figure 4 it would be easier to read if the line types for the disease specific temporal effects were the same as the line types used in the combined temporal effects (make the HIV effect a solid line) and if the shared temporal effect had a third line type to distinguish it.

31. If the shared temporal effect is essentially 0 for 2013 – 2017, it seems like there is minimal evidence that these diseases should be jointly modeled. This should be discussed.

32. It would be more informative to add the credible intervals around the lines in Figure 4. The authors are claiming that HIV risk trend surpassed the TB risk trend after 2014, but this is only based on the point evidence. It’s possible that there is not a statistically significant difference in the temporal effect at any time point.

33. Figure 5 is plotting probabilities, so it makes no sense to have a color category for probability >1 as probabilities can never be bigger than 1. Perhaps consider changing the color scheme for Figures 5 and 6 since in Figures 1-3 you are mapping relative risk and Figures 5-6 are mapping posterior probabilities.

34. In describing Figure 6 the authors claim that the relative risks are compared to the whole of Kenya. Are the other relative risks not compared to Kenya as a whole? Why mention this here? Discussing the standardization method might make this more clear.

Discussion

35. It’s not very clear what the advantage of joint modeling is in this case. Have you tried modeling them separately? I would like a justification for what new was learned through the joint modeling that could not have been learned through modeling the diseases separately.

36. Since the diseases are modeled jointly, it seems like the main thing you would want to make inference on would be the shared parameters as these would tell you if HIV and TB have similar spatiotemporal transmission dynamics. Providing a summary of the statistical significance of the shared parameters could add to this and would be more valid then just looking at the maps and saying they look similar.

37. Line 236. “Joint temporal analysis is important …” It would be nice to have a citation or some rationale on why this is important.

38. Line 245 “Our study successfully detected the spatial congruence….” There is no mention of how the authors went about detecting spatial congruence or what spatial congruence means. It seems like they are talking about finding counties with similar risk, but how would one define “largely similar” mathematically?

39. Line 257 “there was strong evidence of disease-time, disease-space, and space-time interactions” However, there was no reported evidence from the model results. Presumably the authors looked at the credible intervals for these parameters, and those intervals need to be provided for the readers. The maps of the posterior probabilities don’t show this because both the cut points chosen and the coloring are uninformative.

Reviewer #2: In this paper, the authors propose the use of Bayesian hierarchical modelling for joint disease modelling over space and time. A case study in disease mapping of HIV and TB in Kenya is presented to demonstrate the proposed methodology.

Overall, the manuscript is well written, and the methodology is technically sound. In its current form, the presentation of proposed methodology could be improved, including its connections with previous work. Aspects of model presentation and interpretation could also be improved.

The manuscript cites recent studies that have considered spatio-temporal modelling of more than one disease simultaneously. Following this discussion, it is stated that spatio-temporal approaches to joint disease mapping are not common (e.g. line 74). As a reader, it was not clear how modelling multiple related diseases is different to joint disease mapping. Is the proposed methodology an application of one of these recent approaches or is new methodology being proposed? If new methodology is being presented in the manuscript, further clarification of this distinction is required. For example, the paper would benefit from an additional section in the Methods relating the proposed hierarchical model to other methods already published in the literature. This discussion could be aided by explaining differences in the specification of the linear predictor.

Some terms used throughout the manuscript required clarification. For example, what is meant by ‘spatial congruence’ (Abstract, line 31)? As a reader, I was unsure whether this referred to the visualisation of modelled spatial patterns or a specific test being used to measure the degree of spatial correlation.

The presentation of the model equations and subsequent discussion in the Methods section could be improved:

• Line 117: ‘d’ in y_{dst} not defined

• Line 118: suggest adding a second equation here to define the Poisson mean; i.e. u_{dst} = E_{dst}\\rho_{dst}. By defining here, it is easier for the reader to relate \\mu_{dst} with \\rho_{dst} and \\eta_{dst}, supported by text on lines 199-120.

• Lines 121 – 127. This text would be better placed after the presentation of the linear predictors on p7, once the reader has seen the different effects included in the model.

• Line 126: ‘structured effects’. Suggest revising to ‘spatially and/or temporally correlated effects’

• Line 140: which random effects in the proposed model are assumed to be CAR distributed? A sentence linking the assumed neighbourhood structure to relevant prior distributions would help the reader understand its role in the model. This will also help subsequent discussion on lines 158-161.

• Lines 147 – 155: \\theta is not included in the proposed linear predictors at the top of p7. I was therefore unable to connect the discussion of temporal correlation to relevant random effects included in the model (see per previous comment). To improve this discussion, I suggest adding text after line 155 stating which random effects were temporally correlated. Also, on line 155, please define W_{t+}.

• What are the prior distributions for the random effects’ variances/precisions? I could not see any discussion of this in the manuscript. Please provide details of these and brief justification for the choice of prior distributions (i.e. line 171)

• Please provide details on model estimation (assuming MCMC or INLA). Also, how was model convergence assessed?

In the Results section, the presentation of key findings could be improved based on material presented in the Methods section.

• Line 179 (Fig. 1 caption: O_{dst} should be y_{dst} for consistency

• Line 180: what is meant by ‘combined spatial effects’ and how should this combined effect be interpreted? Further explanation of how to interpret different random effects (and combinations of effects) in the Methods section would help the reader better understand the results presented in Figure 2. To appeal to a wider readership, this explanation should be communicated in lay terms where possible.

• Line 185 (disease-specific spatial effects): see previous comment (unsure which effects are being visualised)

• Line 191 (combined temporal effect): see comment about line 180.

Other comments:

• Suggest improving consistent use of terminology throughout the manuscript. For example, ‘spatio-temporal’ versus ‘space-time’; ‘hierarchical Bayesian framework’ versus ‘hierarchical Bayesian model’.

• Lines 82-83: As a reader with no experience in HIV and TB epidemiology, it was unclear what was meant by the phrase ‘co-epidemic overlap thereby exhibiting adverse bidirectional interaction’. Suggest revising this sentence for clarity.

• Line 102: Reference required for national surveillance system.

• Line 141: ‘non-contagious’ should be ‘non-contiguous’

• Data availability: the authors state the all data are fully available without restriction, however I could not find any references to these data in the manuscript.

6. PLOS authors have the option to publish the peer review history of their article (what does this mean?). If published, this will include your full peer review and any attached files.

Reviewer #1: No

Reviewer #2: Yes: Nicole White

---

## [Author Response · Author response to Decision Letter 0]

7 Feb 2020

Editor

Comment 1:

Journal Requirements: When submitting your revision, we need you to address these additional requirements. Please ensure that your manuscript meets PLOS ONE's style requirements, including those for file naming. The PLOS ONE style templates can be found at http://www.journals.plos.org/plosone/s/file?id=wjVg/PLOSOne_formatting_sample_main_body.pdf and http://www.journals.plos.org/plosone/s/file?id=ba62/PLOSOne_formatting_sample_title_authors_affiliations.pdf

Revision:

We have ensured that the revised manuscript meets PLOS ONE's style requirements for the title, authors, affiliations, main body and for file naming

Comment 2:

Please outline in the Methods section how the data was collected, in enough detail for another researcher to reproduce the findings. For instance, please outline the relevant variables for which data was collected. Please also amend your Data availability statement to provide the relevant contact details for where other researchers may apply to access the data. Are you able to provide a list of the counties for which data was collected, in order to ensure reproducibility?

Revision:

We have outlined the data collection processes in the materials and methods section under the data sources sub-section. The relevant variable for this particular study was the number of reported cases per county per year for the period under study. We have amended the data availability statement and provided relevant contact details where other researchers may apply to access the data. Yes, we have provided a list of the counties for which the data was collected as supplementary information S2

Comment 3:

Please include your ethics statement in the Methods section, and clarify whether the ethics committee specifically approved the study

Revision:

We have included the ethics statement in the materials and methods section under the ethical consideration sub-section

Comment 4:

We note that Figures 1-6 in your submission contain map/satellite images which may be copyrighted. All PLOS content is published under the Creative Commons Attribution License (CC BY 4.0), which means that the manuscript, images, and Supporting Information files will be freely available online, and any third party is permitted to access, download, copy, distribute, and use these materials in any way, even commercially, with proper attribution. For these reasons, we cannot publish previously copyrighted maps or satellite images created using proprietary data, such as Google software (Google Maps, Street View, and Earth). For more information, see our copyright guidelines: http://journals.plos.org/plosone/s/licenses-and-copyright. We require you to either (1) present written permission from the copyright holder to publish these figures specifically under the CC BY 4.0 license, or (2) remove the figures from your submission:

Revision:

The shapefiles used in the manuscript for mapping are freely available from a public repository hosted at https://open.africa/dataset/kenya-counties-shapefile and are covered under the creative commons license CC BY 4.0. The images presented in Figures 1-7 are our own generated from the data we used in the study. The data used to generate the maps is owned by NASCOP & NLTP, which are third-party organizations. We have given a statement of the same on the letter to the editor

Comment 5:

We note that you have indicated that data from this study are available upon request. PLOS only allows data to be available upon request if there are legal or ethical restrictions on sharing data publicly. For more information on unacceptable data access restrictions, please see http://journals.plos.org/plosone/s/data-availability#locunacceptable-data-access-restrictions.

b) If there are no restrictions, please upload the minimal anonymized data set necessary to replicate your study findings as either Supporting Information files or to a stable, public repository and provide us with the relevant URLs, DOIs, or accession numbers. For a list of acceptable repositories, please see http://journals.plos.org/plosone/s/dataavailability#loc-recommended-repositories.

Revision:

We have revised the cover letter and indicated that the data used in this study is owned by NASCOP & NLTP which are third-party organizations. There are ethical restrictions in sharing the data imposed by the data owners; we have provided contact information to which data requests may be sent

 

Reviewer 1

Comment 1:

Abstract - First sentence. I would not call spatiotemporal modeling of multiple diseases a “recent extension” because it goes back to at least Knorr-Held and Best, 2001

Revision:

Knorr-Held and Best, 2001 did not combine ideas of spatiotemporal modeling to joint disease mapping, instead they focused a lot on the spatial variation using the shared component model to detect joint and clustering of two diseases. There was no consideration of the temporal variation. In our case we are combining both the spatial and temporal concepts to modeling of multiple diseases, so we have the spatial, temporal, spatiotemporal measures for the two diseases independently and jointly

Comment 2:

Introduction - I agree that the Bayesian structure gives more interpretable estimates, but what makes it useful for mapping multiple diseases is the ability to leverage information between diseases or sub-populations. This point that the robustness of the Bayesian model makes it useful in mapping a single disease for multiple sub-populations or multiple diseases with common risk factors should be discussed more. Explain why it is important to be able to do this, and the benefits of borrowing strength across diseases or sub-populations.

Revision:

We have revised the statement as stated in lines 52-57

Comment 3:

Motivation - Lines 82-85. It would be helpful to have more discussion on similarities between HIV and TB. Jointly modeling both diseases makes strong assumptions about the transmission dynamics and spread of both diseases, and those assumptions are not discussed.

Revision:

We have enriched our "motivation of the study" sub-section as advised by the reviewer. Our emphasis is on jointly modeling HIV & TB on their co-epidemic overlap and not their transmission dynamics.

Comment 4:

Motivation - Line 87. “HIV disease is equally a major global health cause …”. The word such as “concern” should come between the words health and cause.

Revision:

We have updated the statement to read 'HIV disease is equally a global health concern'

Comment 5:

Motivation - Lines 90-91. It seems odd to group transmission dynamics and prevention options together. Expand upon current prevention options that might influence transmission dynamics. Can prevention options be incorporated into the statistical model?

Revision:

a) We have rewritten the statements to make more clearer meaning on our emphasis on the co-epidemic overlap (and not prevention options). We have therefore removed the "prevention options"

b) The scope of this study is not on the prevention options but on the co-epidemic overlap

c) Again, because the prevention options is not our focus on this study, we choose not to incorporate it in the model

Comment 6:

Data sources - More details on the counties in Kenya would be helpful for an international audience. Consider mentioning the population size and geographic size (e.g., median, range).

Revision:

We have included a sub-section titled "study setting" under the material and methods section which provides a description of Kenya interms of geographic placement, population density and GDP among other information. We have also included the geographic presentation and list of counties as supplementary information S2. The county population estimates for the period 2012-2017 are in supplementary information S3 and are also available from the Kenya National Bureau of Statistics (P.O. Box 30266-00100, Nairobi-Kenya; telephone: +254-20-317583; www.knbs.or.ke).

Comment 7:

Data sources - More detail about which people might be included in the case notification data would be helpful. There is some mention of this in the discussion, but it would make more sense to describe who is represented by case notification data when describing and introducing the data.

Revision:

Case notifications are data of positively diagnosed patients seeking treatment and care from health facilities; these are geographically representative of nearby populations. In our study, all the reported cases have been included and serve as a surrogate measure of the entire population of infected persons

Comment 8:

Hierarchical Model - Line 105. The first sentence is unclear as it mentions combining spatio-temporal concepts. In actuality, it is the joint spatio-temporal modeling of two infectious diseases. What is being combined are the underlying spatial and temporal structures. The authors should be more precise with the terminology being used.

Revision:

We are combining the spatio-temporal methods and the modeling of two diseases simultaneously; this is the same as saying joint spatio-temporal modeling of two infectious diseases. What is being combined are the approaches of the underlying spatial & temporal and bivariate disease modeling. We are more precise and consistent with the terminologies used

Comment 9:

Hierarchical Model - Line 108. What does your model borrow from, and how does your model differ from that of [22]? It is stated in the paper that methods are borrowed from [4] and [22] without saying which methods. Log-linear model

Revision:

We have clarified that our model borrows spatial and temporal specifications from the two different studies and [9] and [43]. Lines 167-171 explains specifically how our model differs from that of [9], equally, line 171-172 explains how our model differs from that of [43]

Comment 10:

Log-linear model - Line 117. It should be mentioned that the subscript d stands for disease and to clarify that d=1 is HIV and d=2 is TB or vice-versa.

Revision:

We have mentioned that the subscript d stands for diseases d1=HIV & d2=TB (Line 176-177)

Comment 11:

Log-linear model - Line 120. It would make more sense for the equations with the linear predictors (lines 130 and 131) to appear after the sentence “The linear predictor … family of distributions” because it is easier to follow the model formulation after seeing the equations.

Revision:

We have adjusted accordingly as proposed by the reviewer (Lines 188 - 193)

Comment 12:

Log-linear model - Line 120. How are the expected counts calculated? Knowing this would help in the interpretation of the relative risk.

Revision:

Yes we have elaborated on the calculation of the offset - expected counts (Lines 180 - 187)

Comment 13:

Log-linear model - Lines 122-123. When describing the model formulation, instead of writing “specific patterns” or “specific random effects”, it would be clearer to use the phrase “disease-specific patterns” or “disease-specific effects”

Revision:

We have corrected so that the "specific patterns" reads "disease-specific patterns" and "specific random effects" reads "disease-specific random effects"

Comment 14:

Log-linear model - Line 126. Talking about random effects is not standard in the Bayesian framework because all parameters have distributions and are thus considered to be “random.”

Revision:

There are cases where we have fixed effects so it becomes important to indicate they are random effects. Equally in the Bayesian framework, the term random effects clarifies that a parameter is stochastic and not static

Comment 15:

Log-linear model - When discussing the model formulation, it would be helpful to give the symbols for each effect specified. For example, “the disease-specific effects, α¬d, β¬ds, and γdt allowed for departures …” Consider moving the first paragraph of the Bayesian model specification right after introducing the model. Furthermore, the spatial correlation and temporal correlation sections introduce parameters that are not part of the model. Because of that, the paper does not flow and it is hard to follow what parameters have what distributions. Consider restructuring to make it easier to follow.

Revision:

We have restructured the discussion of the model formulation to give the paper a better flow and easy read

Comment 16:

Log-linear model - It is a very strong assumption to have two diseases share the δ and κ terms where one model includes the reciprocal. More discussion and justification of this formulation is needed.

Revision:

We have provided more discussion and justification on the formulation of δ and κ terms. Refer to lines 199 - 204 and lines 226-228

Comment 17:

Log-linear model - Be specific what is meant by structured and unstructured effects. Presumably the structured effects have the ICAR model and the unstructured effects have independent normal priors, but this should be explicitly stated.

Revision:

Both the structured and unstructured spatial effects assume the ICAR prior whereas the structured and unstructured temporal effects assume an RW 1 prior

Comment 18:

Bayesian Model Specification - Line 157 is confusing. Please rephrase.

Revision:

We have rewritten the entire section and removed the statement to ensure clarity

Comment 19:

Bayesian Model Specification - Following from comment 15, it would make more sense to move the part of the Bayesian model specification section discussing the spatial effects to the spatial correlation section and the part discussing the temporal effects to the temporal correlation section

Revision:

We have moved the sections as advised

Comment 20:

Bayesian Model Specification - When the ICAR weights in the spatial correlation section are introduced, the S notation is used in the neighbourhood matrix. In the Bayesian model specification section, the W notation is used. Notation should be consistent

Revision:

We have ensured the notations are consistent. The W are the weights in the correlation. S is the subscript for the spatial parameters defining the geographically adjacent areas

Comment 21:

Bayesian Model Specification - The Bayesian model specification section seems incomplete. There is no discussion of prior distributions for the α parameters or for the hyperparameters (τλ, τβ, τξ, τγ, τν), no discussion of convergence assessment, and no discussion of computation. It looks like from the code that the authors used INLA - this should be mentioned in the manuscript

Revision:

We have discussed on the specifications of prior distributions for the α parameters and for the hyperparameters (τλ, τβ, τξ, τγ, τν) using INLA default. We used INLA for the model estimations. INLA potentially provides a major advance in ability to handle complex models with large predictor spaces. The Laplace Approximation (LA) in INLA does not require convergence checking (unlike McMC)

Comment 22:

Bayesian Model Specification - It is not clear what the hyperparameters for the spatial effects are. Consider adding the ICAR distribution to the spatial correlation section to identify that these hyperparameters come from the variance of the ICAR model

Revision:

We have rewritten the entire section on the prior specifications, now it is much clearer

Comment 23:

Bayesian Model Specification - When talking of the hyperparameters of the temporal effects, it is confusing because you used φ in the temporal correlation section, but then switched to τ in the Bayesian model specification section. Notation should be consistent

Revision:

We have rewritten the entire section on the prior specifications and ensured that the notations are consistent

Comment 24:

Results - All the maps need more descriptive labels – it is impossible to know which part of Fig 1 is HIV or TB. And without reading the text you wouldn’t know the maps are on the relative risk scale because they that are labelled as the “effect” not the exponential of the effect

Revision:

We have ensured the maps have descriptive labels as proposed. The term effects is applied on either the spatial patterns or temporal trends and not on their relative risks. We have replaced the term effects with either patterns or trends.

Comment 25:

Results - The switch from relative risk to the term standardized incidence ratio seems unnecessary and without explanation. It would be more clear to state that you are plotting exp(η) as the relative risk

Revision:

We have clarified on the relative risks and the standardized incidence ratios. Equally, we have indicated where we have plotted the relative risks and where we have plotted the SIRs

Comment 26:

Results - The color scale for the maps seems very arbitrary. Relative risk is a ratio – so therefore a 0.71 – 0.99 decrease is comparable to a 1.01 - 1.4 increase. Coloring all relative risks above 1 the same is extremely misleading and uninformative. Relative risks of 1 just mean the risk is what you would expect, it doesn’t make sense for this to be the darkest red

Revision:

Our intention was to generate maps and use them to identify elevated risk areas with relative risks above 1. That explains the sequential coloring to depict counties of low risk (light color intensity) to counties of high risk (deep color intensity). Relative risks above one are indicative of elevated risk areas that require intervention.

Comment 27:

Results - The authors state “the progression of the risk during the period 2012 – 2017 was much faster for TB as compared to HIV” based only on looking at the maps. You can use the posterior distributions of γ1t and γ2t to compute the posterior probability that the progression of risk for TB is faster than the progression of risk for HIV. This would be much more informative

Revision:

The values generated on the maps are posterior probabilities which are better estimates compared to the SIRs

Comment 28:

Results - The labels for figure 2 and 3 say posterior means of each effects are plotted, but they are actually plotting those effects on the relative risk scale (exponentiated). The labels should indicate this.

Revision:

We have indicated the relative risks as suggested by the reviewer

Comment 29:

Results - Indicate on Figure 4 that plots show exponentiated values by labelling the y-axis. It is confusing what is being plotted as the combined temporal effect, because for both HIV and TB in 2012 the shared effect is above 1 and the combined effect is less than the disease specific effects for both diseases, but the claim is that this is plotting the sum? Looking in the R code it does not appear that the authors plotted the correct thing either for the shared effect or for the sum of the shared and disease specific effects.

Revision:

We have labelled the y-axis accordingly. The temporal scaling parameter on the shared term is depicting the joint overall temporal relative risk trend. The contribution of the shared component to the overall relative risk is weighted by the scaling parameter to allow different risk gradients for each disease and that explains why the temporal combined effect for each disease is less than the shared temporal effect and the disease-specific temporal effect. The plots are correct

Comment 30:

Results - In Figure 4 it would be easier to read if the line types for the disease specific temporal effects were the same as the line types used in the combined temporal effects (make the HIV effect a solid line) and if the shared temporal effect had a third line type to distinguish it.

Revision:

We have adjusted based on the reviewer's suggestions

Comment 31:

Results - If the shared temporal effect is essentially 0 for 2013 – 2017, it seems like there is minimal evidence that these diseases should be jointly modelled. This should be discussed

Revision:

We have discussed this further; the specific temporal trends show a strong departure from the shared temporal pattern. The shared temporal effect captures the overall decreasing trend in time with estimates close to zero for most years. Even though the shared effect shows minimal evidence for joint modeling, the disease specific and combined temporal effects show very similar temporal pattern. We have noted and discussed this output further as a concern especially for integrated interventions towards suppressing the spread of the two disease. Refer to lines 305-313

Comment 32:

Results - It would be more informative to add the credible intervals around the lines in Figure 4. The authors are claiming that HIV risk trend surpassed the TB risk trend after 2014, but this is only based on the point evidence. It’s possible that there is not a statistically significant difference in the temporal effect at any time point.

Revision:

We have included Table 1 which shows the summary statistics of the shared and disease specific spatial and temporal effects

Comment 33:

Results - Figure 5 is plotting probabilities, so it makes no sense to have a color category for probability >1 as probabilities can never be bigger than 1. Perhaps consider changing the color scheme for Figures 5 and 6 since in Figures 1-3 you are mapping relative risk and Figures 5-6 are mapping posterior probabilities

Revision:

We have corrected the range as advised by the reviewer

Comment 34:

Results - In describing Figure 6 the authors claim that the relative risks are compared to the whole of Kenya. Are the other relative risks not compared to Kenya as a whole? Why mention this here? Discussing the standardization method might make this more clear

Revision:

We have rephrased the statement and it is much clearer refer to line 272 and figure 7

Comment 35:

Discussion - It’s not very clear what the advantage of joint modeling is in this case. Have you tried modeling them separately? I would like a justification for what new was learned through the joint modeling that could not have been learned through modeling the diseases separately

Revision:

The disease specific and combined parameters have taken care of modeling the diseases separately and jointly respectively. We have provided a justification for new lessons for joint modeling which could not be learnt through modeling the diseases separately.

Comment 36:

Discussion - Since the diseases are modelled jointly, it seems like the main thing you would want to make inference on would be the shared parameters as these would tell you if HIV and TB have similar spatiotemporal transmission dynamics. Providing a summary of the statistical significance of the shared parameters could add to this and would be more valid then just looking at the maps and saying they look similar.

Revision:

The posterior probability maps and spatial patterns depicting measures of uncertainty of the spatial effects are presented in figure 7, these are equally valid statistical measures presented in a map

Comment 37:

Discussion - Line 236. “Joint temporal analysis is important …” It would be nice to have a citation or some rationale on why this is important.

Revision:

We have provided a citation and rationale on why joint temporal analysis is important (Line 306)

Comment 38:

Discussion - Line 245 “Our study successfully detected the spatial congruence….” There is no mention of how the authors went about detecting spatial congruence or what spatial congruence means. It seems like they are talking about finding counties with similar risk, but how would one define “largely similar” mathematically?

Revision:

For this particular study, spatial congruence referred to the visualization of the modelled spatial patterns using the posterior means and relative risks which are statistically significant and reliable estimates

Comment 39:

Discussion - Line 257 “there was strong evidence of disease-time, disease-space, and space-time interactions” However, there was no reported evidence from the model results. Presumably the authors looked at the credible intervals for these parameters, and those intervals need to be provided for the readers. The maps of the posterior probabilities don’t show this because both the cut points chosen and the colouring are uninformative.

Revision:

We have reported and provided the credible intervals in table 1

 

Reviewer 2

Comment 1:

Overall, the manuscript is well written, and the methodology is technically sound. In its current form, the presentation of proposed methodology could be improved, including its connections with previous work. Aspects of model presentation and interpretation could also be improved. The manuscript cites recent studies that have considered spatio-temporal modelling of more than one disease simultaneously. Following this discussion, it is stated that spatio-temporal approaches to joint disease mapping are not common (e.g. line 74). As a reader, it was not clear how modelling multiple related diseases is different to joint disease mapping. Is the proposed methodology an application of one of these recent approaches or is new methodology being proposed? If new methodology is being presented in the manuscript, further clarification of this distinction is required. For example, the paper would benefit from an additional section in the Methods relating the proposed hierarchical model to other methods already published in the literature. This discussion could be aided by explaining differences in the specification of the linear predictor.

Revision

a) We have clarified what we meant.

Modeling multiple related is the same as joint disease mapping. To maintain consistency, we are using "modeling multiple diseases simultaneously". There are two concepts here, the first is the spatio-temporal and the second in the modeling of multiple disease simultaneously.

The manuscript has discussed the evolution of disease mapping from the recent studies cited

* the first one is the purely spatial analysis of a single disease

* the second is the inclusion of the time dimension offering additional benefits over purely spatial analysis by allowing for the concurrent study of persistent and unusual spatial patterns over time

* the third is the joint spatial modeling without the temporal effect - this allows for modeling of multiple related diseases with common risk factors. It outspreads the standard univariate disease mapping methodologies

* the fourth is the combination of the spatiotemporal approaches to the modeling of multiple diseases simultaneously

The focus of this study is on this fourth concept which has not been studied as much

b) The methodology itself is an application of recent approaches and not necessarily novel, but the application of the model to HIV with TB in Kenya is new

c) Since we are using an already existing application, we need not to compare the specifications of the linear predictors

Comment 2:

Some terms used throughout the manuscript required clarification. For example, what is meant by ‘spatial congruence’ (Abstract, line 31)? As a reader, I was unsure whether this referred to the visualization of modelled spatial patterns or a specific test being used to measure the degree of spatial correlation.

Revision

For this particular study, spatial congruence referred to the visualization of the modelled spatial patterns

Comment 3:

The presentation of the model equations and subsequent discussion in the Methods section could be improved:

Line 117: ‘d’ in y_{dst} not defined

Revision:

We have improved on the presentation of the model equations and the subsequent discussions in the methods section

We have mentioned that the subscript d stands for diseases d1=HIV & d2=TB

Comment 4:

Line 118: suggest adding a second equation here to define the Poisson mean; i.e. u_{dst} = E_{dst}\\rho_{dst}. By defining here, it is easier for the reader to relate \\mu_{dst} with \\rho_{dst} and \\eta_{dst}, supported by text on lines 199-120

Revision:

We have added a second equation to define the poisson mean. We have also elaborated on the calculation of the offset - expected counts

Comment 5:

Lines 121 – 127. This text would be better placed after the presentation of the linear predictors on p7, once the reader has seen the different effects included in the model.

Revision:

We have moved the text accordingly

Comment 6:

Line 126: “structured effects”. Suggest revising to ‘spatially and/or temporally correlated effects’

Revision:

In our model, we do have shared spatially and temporally correlated random effects which we have assigned to be the spatial and temporal structured effects

Comment 7:

Line 140: which random effects in the proposed model are assumed to be CAR distributed? A sentence linking the assumed neighbourhood structure to relevant prior distributions would help the reader understand its role in the model. This will also help subsequent discussion on lines 158-161.

Revision:

The spatial random effects. The equation in line 220 is an elaboration of the preceding sentence (line 218-219) on how we defined the correlation for geographically adjacent areas. We have updated accordingly

Comment 8:

Lines 147 – 155: \\theta is not included in the proposed linear predictors at the top of p7. I was therefore unable to connect the discussion of temporal correlation to relevant random effects included in the model (see per previous comment). To improve this discussion, I suggest adding text after line 155 stating which random effects were temporally correlated. Also, on line 155, please define W_{t+}.

Revision:

We have rewritten the entire section on the prior specifications and ensured that the notations are consistent

Comment 9:

What are the prior distributions for the random effects’ variances/precisions? I could not see any discussion of this in the manuscript. Please provide details of these and brief justification for the choice of prior distributions (i.e. line 171)

Revision:

We have elaborated on the prior distributions for the random effects’ variances/precisions and provided details of these and brief justification for the choice of prior distributions

Comment 10:

Please provide details on model estimation (assuming MCMC or INLA). Also, how was model convergence assessed?

Revision:

We used INLA for the model estimations. INLA potentially provides a major advance in ability to handle complex models with large predictor spaces. The Laplace Approximation (LA) in INLA does not require convergence checking (unlike McMC)

Comment 11:

In the Results section, the presentation of key findings could be improved based on material presented in the Methods section.

Line 179 (Fig. 1 caption: O_{dst} should be y_{dst} for consistency

Revision:

We have improved the presentation of the findings to match the methodology presented. We have ensured consistency in the parameter symbols

Comment 12:

Line 180: what is meant by ‘combined spatial effects’ and how should this combined effect be interpreted? Further explanation of how to interpret different random effects (and combinations of effects) in the Methods section would help the reader better understand the results presented in Figure 2. To appeal to a wider readership, this explanation should be communicated in lay terms where possible.

Revision:

The combined spatial effects are the "shared spatial effects in relation to TB (or HIV) + TB (or HIV) specific spatial effects”.

Comment 13:

Line 185 (disease-specific spatial effects): see previous comment (unsure which effects are being visualised)

Revision:

Disease specific spatial effects are the spatial effects specific to a given disease. We are visualizing the shared, disease-specific, and combined effects

Comment 14:

Line 191 (combined temporal effect): see comment about line 180.

Revision:

The 'combined temporal effects' are the 'shared temporal effects in relation to TB (or HIV) + TB (or HIV) specific temporal effects.

Comment 15:

Suggest improving consistent use of terminology throughout the manuscript. For example, ‘spatio-temporal’ versus ‘space-time’; ‘hierarchical Bayesian framework’ versus ‘hierarchical Bayesian model’.

Revision:

We have improved the consistency of the terminologies as proposed. So, we use the 'spatio-temporal' and 'hierarchical Bayesian model' consistently through the study

Comment 16:

Lines 82-83: As a reader with no experience in HIV and TB epidemiology, it was unclear what was meant by the phrase ‘co-epidemic overlap thereby exhibiting adverse bidirectional interaction’. Suggest revising this sentence for clarity.

Revision:

What we mean by the ‘co-epidemic overlap thereby exhibiting adverse bidirectional interaction' is that HIV increases the risk of TB infection and TB slows down CD4 recovery and increases progression to AIDS and death among the HIV infected

Comment 17:

Line 102: Reference required for national surveillance system

Revision:

We have provided the reference for the national surveillance system in the data availability statement and in the data sources sub-section of the materials and methods section

Comment 18:

Line 141: ‘non-contagious’ should be ‘non-contiguous’

Revision:

We have adjusted to read 'non-contiguous'

Comment 19:

Data availability: the authors state the all data are fully available without restriction, however I could not find any references to these data in the manuscript

Revision:

We have amended your Data availability statement to provide the relevant contact details for where other researchers may apply to access the data.

Data availability statement

The HIV and TB data used in this study are available from NASCOP (P.O. Box 19361-00202, Nairobi-Kenya; telephone: +254-775597297; info@nascop.or.ke; https://dwh.nascop.org/) and NLTP (P.O. Box 20781-00202, Nairobi-Kenya; telephone: +254-773977440; info@nltp.co.ke; http://pms.dltld.or.ke/). Interested researchers may contact NASCOP (head@nascop.or.ke) for the HIV data and NLTP (head@nltp.co.ke) for the TB data. The county population estimates for the period 2012-2017 are available from the Keya National Bureau of Statistics (P.O. Box 30266-00100, Nairobi-Kenya; telephone: +254-20-317583; www.knbs.or.ke).

---

## [Decision Letter · Decision Letter 1]

25 Mar 2020

PONE-D-19-25050R1

Bayesian hierarchical modeling of joint spatiotemporal risk patterns for Human Immunodeficiency Virus (HIV) and Tuberculosis (TB) in Kenya

PLOS ONE

Dear Ms Otiende,

Thank you for submitting your manuscript to PLOS ONE. After careful consideration, we feel that it has merit but does not fully meet PLOS ONE’s publication criteria as it currently stands. Therefore, we invite you to submit a revised version of the manuscript that addresses the points raised during the review process.

Please address all of the comments from both reviewers. In particular, note that one reviewer has commented that the manuscript's conclusions and interpretations remain unclear and the figures remain misleading. The reviewer also comments that the results section is difficult to follow and it is still unclear whether the conclusions are supported by the data. These issues must be addressed.

We would appreciate receiving your revised manuscript by May 09 2020 11:59PM. To enhance the reproducibility of your results, we recommend that if applicable you deposit your laboratory protocols in protocols.io, where a protocol can be assigned its own identifier (DOI) such that it can be cited independently in the future. For instructions see: http://journals.plos.org/plosone/s/submission-guidelines#loc-laboratory-protocols

We look forward to receiving your revised manuscript.

Kind regards,

Eric Forgoston

Academic Editor

PLOS ONE

Reviewers' comments:

Reviewer's Responses to Questions

**Comments to the Author**

1. If the authors have adequately addressed your comments raised in a previous round of review and you feel that this manuscript is now acceptable for publication, you may indicate that here to bypass the “Comments to the Author” section, enter your conflict of interest statement in the “Confidential to Editor” section, and submit your "Accept" recommendation.

Reviewer #1: (No Response)

Reviewer #2: All comments have been addressed

2. Is the manuscript technically sound, and do the data support the conclusions?

Reviewer #1: Partly

Reviewer #2: Yes

3. Has the statistical analysis been performed appropriately and rigorously? 

Reviewer #1: No

Reviewer #2: Yes

4. Have the authors made all data underlying the findings in their manuscript fully available?

Reviewer #1: Yes

Reviewer #2: Yes

5. Is the manuscript presented in an intelligible fashion and written in standard English?

Reviewer #1: Yes

Reviewer #2: Yes

6. Review Comments to the Author

Reviewer #1: SUMMARY/OVERALL COMMENTS

Overall, the authors followed some of the reviewer comments in the data description and methodology sections to improve this version of the manuscript. However, the conclusions and interpretations are still unclear and the figures presented remain misleading. In the results section, many of the plots claim to be showing relative risk, but in the text of the document it is not described how those relative risk values are calculated, and the figure labels do not clarify. Plots of the overall estimated relative risk in each region over time should be shown. Plots of relative risk need to have a different color scheme than currently shown since the current scheme is not informative. More description of what shared, disease-specific, and combined spatial and temporal trends/patterns are in relation to the model parameters needs to be added in the text and not just in the figure labels. As currently presented, the results section is difficult to follow and it remains unclear if the conclusions are supported by the data.

SPECIFIC COMMENTS:

Abstract

1. The first sentence should be deleted. It begins with “the spatiotemporal modeling of multiple diseases simultaneously” and ends by reordering those same words as “space-time analysis to model multiple diseases simultaneously.” The Abstract can begin with “The simultaneous spatiotemporal modeling of multiple related diseases strengthens inferences by borrowing information between related diseases.”

Introduction

2. The sentence on lines 50-51 can be deleted. “The latest and most recent extension is the joint spatiotemporal modeling of multiple diseases.” The sentence does not add to the discussion in that paragraph and “latest and most recent” is not necessarily true as the authors cite in the subsequent paragraph papers that do joint spatiotemporal modeling going back to 2004, 2006, and 2008.

3. Line 81. Change “These model maps” to “Maps generated from these models”.

Motivation

4. Line 103. “provides” should just be “provide”.

5. The added discussion about the relationship between HIV and TB helps address why one might jointly model these diseases, however it would benefit from a description of the raw data. A data summary or a map of the raw data for example.

6. Lines 99-100. The causality inferred by this statement is not clear. Surely one disease does not cause the other, but it can make one more susceptible to the other. This should be stated more clearly.

Study location

7. Line 121-122. Change “The population density has remained on the increase…” to “The population density has continued to increase…”

Hierarchical Model specification

8. Line 164-165. Change to “We formulated a statistical model that applied spatiotemporal methods to the modeling of HIV and TB simultaneously.” This phrasing more accurately describes what the authors are doing which is using existing methodology in a new setting.

9. Line 167 and 168. Change the word “mapped” to “modeled” in both instances. Model results are what are being shown in a map.

Log-linear model

10. Line 181. Although the elaboration improves the paper, some aspects of the description and notation used for the standardization process remain unclear. Is N the same for each county? I would assume not, but that point is not entirely clear in the way that it is described. The current notation N=P/T implies that P and N are the same for each county. Indices using s and t could help clarify this point to know if N changes per county and/or over time.

11. Line 184. Technically P_{dst} would not need d in this instance, but it could in general be there if the two diseases had different susceptible populations.

12. It seems like the shared spatial and the disease specific spatial terms being in the same model may not all be identifiable individually. The figures seem to suggest this as well. In the Richardson et al paper that the methods are based upon, there is only a beta_{2s} parameter and the beta_{1s} is the reference category as specified on pages 392 and 393 of their paper. This should be addressed.

Bayesian Model Specification

13. Line 223. Please write out how the weighted matrix Q is used in the random walk of order 1.

14. Lines 225-226. Please specify whether the normal distribution is being parameterized in terms of the variance or precision. The fact that alpha has a value of 0 makes it seem like this is a precision parameter, while the scaling parameters are referenced as sigma^2 which makes it seem like a variance. Also, what value was used for sigma^2 in the prior?

15. Line 226. Why are the scaling parameters not restricted to be positive? If I am understanding correctly, they cannot equal 0 since you use 1/delta, but the prior is centered at 0 which is confusing.

Results

16. Lines 245-246. It is reported that there is statistical significance at the 5% level as depicted by the credible intervals. Is this for both the intercept (alpha) and the precision (tau) parameters? Please include the delta and kappa parameters as well. The precision terms should not be include 0 anyway since that is a boundary. It would be informative to talk about the size of the precision terms. For instance, it is interesting that the shared component parameters for beta have larger precision than lambda does. This would suggest that there is very little variance in the shared parameter relative to the disease specific parameter.

17. Figure 5. This plot could be improved by adding error bars for the 95% credible intervals. The authors are interpreting that the combined temporal trend plot of HIV risk is lower in 2012 and 2013, but 2015 and after it is TB that has higher risk. Error bars that do or do not overlap would better demonstrate the risk difference.

18. Figure 6. The red category is for only a probability equal to 1. There are no regions showing this red color. It would be more informative to have the ranges as 0-0.25, 0.26-0.50, 0.51-0.75, 0.76-1.00.

19. Figure 7. The upper bound of the legend is 1.00 in the left panel. Is this correct since panel a is supposedly showing a relative risk? In the figure on the right, there is a typo in the lowest probability values. What is the justification for breaking probability into three unequal sized bins?

20. Lines 269-270. Figure 6 is interpreted as “over the years the joint incidence risk reduced progressively with all spatial regions having a relative risk less than 1.” Figure 6 is showing probabilities, not relative risk, so this interpretation is wrong.

21. Lines 272-273. Why are there disease specific parameters in the joint disease risk?

22. I suggest changing the color schemes between the RR maps and the probability maps to avoid confusion.

Discussion

23. It is unclear what is being plotted in Figure 7 because it claims to be joint disease risk, but the label includes disease specific parameters. To address the significance of the shared parameters, you should look at posterior probabilities of lambda and xi (the shared parameters) being different from 0.

24. How are relative risks determined to be statistically significant? If congruence is purely visual it should be interpreted as descriptive. If congruence is based on posterior means there should be some predetermined threshold and posterior means that differ by less than the threshold would be defined as congruent.

Reviewer #2: The revised manuscript is much improved and has sufficiently addressed my comments.

Minor comments:

Line 226 - typo in prior distribution for the intercept N(0,0). Please correct value for variance

Line 249 - additional details needed in Table 1 caption. Please define interpretation of each column e.g. 2.5th = 2.5th percentile of posterior distribution.

Figure 5 - can 95% credible intervals be added to this figure? This would provide the reader with further information regarding uncertainty about temporal trends.

The scales for Figures 6 and 7(different posterior probabilities) are not the same. Changing these to a consistent scale would improve overall presentation.

7. PLOS authors have the option to publish the peer review history of their article (what does this mean?). If published, this will include your full peer review and any attached files.

Reviewer #1: No

Reviewer #2: Yes: Nicole White

---

## [Author Response · Author response to Decision Letter 1]

17 May 2020

Reviewer 1

Overview:

Overall, the authors followed some of the reviewer comments in the data description and methodology sections to improve this version of the manuscript. However, the conclusions and interpretations are still unclear and the figures presented remain misleading. In the results section, many of the plots claim to be showing relative risk, but in the text of the document it is not described how those relative risk values are calculated, and the figure labels do not clarify. Plots of the overall estimated relative risk in each region over time should be shown. Plots of relative risk need to have a different color scheme than currently shown since the current scheme is not informative. More description of what shared, disease-specific, and combined spatial and temporal trends/patterns are in relation to the model parameters needs to be added in the text and not just in the figure labels. As currently presented, the results section is difficult to follow and it remains unclear if the conclusions are supported by the data.

Revision

The script has been revised accordingly to address the reviewer’s comments.

Comment 1:

The first sentence should be deleted. It begins with “the spatiotemporal modeling of multiple diseases simultaneously” and ends by reordering those same words as “space-time analysis to model multiple diseases simultaneously.” The Abstract can begin with “The simultaneous spatiotemporal modeling of multiple related diseases strengthens inferences by borrowing information between related diseases.”

Revision:

[Lines 18 - 19]: The first sentence adopts the recommendation from the reviewer. 

Comment 2:

The sentence on lines 50-51 can be deleted. “The latest and most recent extension is the joint spatiotemporal modeling of multiple diseases.” The sentence does not add to the discussion in that paragraph and “latest and most recent” is not necessarily true as the authors cite in the subsequent paragraph papers that do joint spatiotemporal modeling going back to 2004, 2006, and 2008.

Revision:

[Line 51]: The statement "latest and most recent extension …" has been replaced with "another extension … "

Comment 3:

Line 81. Change “These model maps” to “Maps generated from these models”.

Revision:

[Lines 81 - 82]: Adopted the reviewer's suggestion

Comment 4:

Line 103. “provides” should just be “provide”.

Revision:

[Line 103]: Adopted the reviewer's suggestion

Comment 5:

The added discussion about the relationship between HIV and TB helps address why one might jointly model these diseases, however it would benefit from a description of the raw data. A data summary or a map of the raw data for example.

Revision:

[Lines 100 - 108] We have included some discussion on the TB - HIV situation in Kenya as observed by previous researchers as a motivation for further studying these two diseases. 

[Lines 164 - 168] We have also included another section under the methods section after the data source sub-section to describe the variables in the study: HIV observed, TB observed, and the population estimates per county per year

Comment 6:

Lines 99-100. The causality inferred by this statement is not clear. Surely one disease does not cause the other, but it can make one more susceptible to the other. This should be stated more clearly.

Revision:

[Lines 99 - 101]: The statement has been stated more clearly

Comment 7:

Line 121-122. Change “The population density has remained on the increase…” to “The population density has continued to increase…”

Revision:

[Line 122 - 123]: Adopted the reviewer's suggestion

Comment 8:

Line 164-165. Change to “We formulated a statistical model that applied spatiotemporal methods to the modeling of HIV and TB simultaneously.” This phrasing more accurately describes what the authors are doing which is using existing methodology in a new setting.

Revision:

[Line 165]: Adopted the reviewer's suggestion

Comment 9:

Line 167 and 168. Change the word “mapped” to “modeled” in both instances. Model results are what are being shown in a map.

Revision:

[Lines 168 - 169]: Adopted the reviewer's suggestion

Comment 10:

Line 181. Although the elaboration improves the paper, some aspects of the description and notation used for the standardization process remain unclear. Is N the same for each county? I would assume not, but that point is not entirely clear in the way that it is described. The current notation N=P/T implies that P and N are the same for each county. Indices using s and t could help clarify this point to know if N changes per county and/or over time.

Revision:

[Line 182]: The notation and descriptions used for the standardization process are clarified.

Comment 11:

Line 184. Technically P_{dst} would not need d in this instance, but it could in general be there if the two diseases had different susceptible populations.

Revision:

[Line 185]: Adopted the reviewer's suggestion

Comment 12:

It seems like the shared spatial and the disease specific spatial terms being in the same model may not all be identifiable individually. The figures seem to suggest this as well. In the Richardson et al paper that the methods are based upon, there is only a beta_{2s} parameter and the beta_{1s} is the reference category as specified on pages 392 and 393 of their paper. This should be addressed.

Revision:

Richardson et al use an asymmetric formulation for their model as their interest was to determine the differential risk patterns, that’s the reason they used the beta_{i} as the female differential from the shared component. In our paper, we are using the shared component approach to separate the underlying risk of each disease into a shared component (common to both diseases) and disease-specific component for each disease. The estimated spatial patterns presented for the shared and disease-specific components are distinct and show no sign of identifiability problem. We imposed a sum-to-zero constraints on both the spatial and temporal random effects to minimize any identifiability problem on the intercept. Sum-to-zero constraints are considered as guaranteeing model identifiability.

Comment 13:

Line 223. Please write out how the weighted matrix Q is used in the random walk of order 1.

Revision:

[Lines 240 - 251]: Adopted the reviewer's suggestion

Comment 14:

Lines 225-226. Please specify whether the normal distribution is being parameterized in terms of the variance or precision. The fact that alpha has a value of 0 makes it seem like this is a precision parameter, while the scaling parameters are referenced as sigma^2 which makes it seem like a variance. Also, what value was used for sigma^2 in the prior?

Revision:

[Lines 252 - 253] * We have specified the normal distribution is parameterized in terms of the variance, we have also expressed the precision parameter and explained the parameters correctly. We have also defined improper flat priors for the alpha which means it is not a probability distribution on its own but yields a well-defined posterior.

Comment 15:

Line 226. Why are the scaling parameters not restricted to be positive? If I am understanding correctly, they cannot equal 0 since you use 1/delta, but the prior is centered at 0 which is confusing.

Revision:

[Lines 254 – 259] We have restricted the scaling parameters to positive by including the expression delta > 0 and kappa > 0. Kappa and delta are symmetric around zero on their log-scale

Comment 16:

Lines 245-246. It is reported that there is statistical significance at the 5% level as depicted by the credible intervals. Is this for both the intercept (alpha) and the precision (tau) parameters? Please include the delta and kappa parameters as well. The precision terms should not be include 0 anyway since that is a boundary. It would be informative to talk about the size of the precision terms. For instance, it is interesting that the shared component parameters for beta have larger precision than lambda does. This would suggest that there is very little variance in the shared parameter relative to the disease specific parameter.

Revision:

[Lines 275 - 283] We have adjusted our explanation accordingly. For the intercepts (alpha) - it is evident that the values are significantly different from zero (as depicted by the credible intervals) implying that the overally (nationwide) disease-specific relative risk is above 1, which implies the country is general is still at a high risk for both infections. For the precision (tau) parameters, we have presented the variance measures instead of the precision values. The percentage of the variation expresses what each of the parameters contributes to the explained variability of HIV and TB. From this we are able to see that the shared components explain the most variability (comparative variation in the shared parameter relative to the disease specific parameters). Both the shared spatial and components dominate the model and cannot be left out of the model.

Comment 17:

Figure 5. This plot could be improved by adding error bars for the 95% credible intervals. The authors are interpreting that the combined temporal trend plot of HIV risk is lower in 2012 and 2013, but 2015 and after it is TB that has higher risk. Error bars that do or do not overlap would better demonstrate the risk difference.

Revision:

Figure 5: we have adopted the suggestion from the reviewer

Comment 18:

Figure 6. The red category is for only a probability equal to 1. There are no regions showing this red color. It would be more informative to have the ranges as 0-0.25, 0.26-0.50, 0.51-0.75, 0.76-1.00.

Revision:

Figure 6: We have adopted the suggestion from the reviewer. We also wish to emphasize that the figure is designed for a comparative illustration. The main information that authors were seeking to showcase was a transition of joint infection across Kenya. The authors were keen to present the spatiotemporal trajectory of the joint risk

Comment 19:

Figure 7. The upper bound of the legend is 1.00 in the left panel. Is this correct since panel a is supposedly showing a relative risk? In the figure on the right, there is a typo in the lowest probability values. What is the justification for breaking probability into three unequal sized bins?

Revision:

We have removed figure seven because it is confusing

Comment 20:

Lines 269-270. Figure 6 is interpreted as “over the years the joint incidence risk reduced progressively with all spatial regions having a relative risk less than 1.” Figure 6 is showing probabilities, not relative risk, so this interpretation is wrong.

Revision:

[Line 306 - 308] We have adjusted to interpret the probabilities and not the relative risk

Comment 21:

Lines 272-273. Why are there disease specific parameters in the joint disease risk?

Revision:

We have removed figure seven because it is confusing

Comment 22:

I suggest changing the color schemes between the RR maps and the probability maps to avoid confusion.

Revision:

Adopted the reviewer’s suggestion.

Comment 23:

It is unclear what is being plotted in Figure 7 because it claims to be joint disease risk, but the label includes disease specific parameters. To address the significance of the shared parameters, you should look at posterior probabilities of lambda and xi (the shared parameters) being different from 0.

Revision:

We have removed figure seven because it is confusing

Comment 24:

How are relative risks determined to be statistically significant? If congruence is purely visual it should be interpreted as descriptive. If congruence is based on posterior means there should be some predetermined threshold and posterior means that differ by less than the threshold would be defined as congruent.

Revision:

Our description for spatial congruence is areas of HIV high risks were found to be also high-risk areas for TB (converse is not true as some high-risk areas of TB were not high-risk areas for HIV). Yes, our congruence is purely visual so we changed the term from congruence to similarity.

 

Reviewer 2

Comment 1:

The revised manuscript is much improved and has sufficiently addressed my comments. Line 226 - typo in prior distribution for the intercept N(0,0). Please correct value for variance

Revision

[Line 253] we have defined an improper flat priors for the intercept

Comment 2:

Line 249 - additional details needed in Table 1 caption. Please define interpretation of each column e.g. 2.5th = 2.5th percentile of posterior distribution.

Revision

[Line 282] we have amended as proposed by the reviewer

Comment 3:

Figure 5 - can 95% credible intervals be added to this figure? This would provide the reader with further information regarding uncertainty about temporal trends.

Revision:

Figure 5: we have adopted the suggestion from the reviewer

Comment 4:

The scales for Figures 6 and 7(different posterior probabilities) are not the same. Changing these to a consistent scale would improve overall presentation.

Revision:

We have removed figure seven because it is confusing.

---

## [Decision Letter · Decision Letter 2]

28 May 2020

Bayesian hierarchical modeling of joint spatiotemporal risk patterns for Human Immunodeficiency Virus (HIV) and Tuberculosis (TB) in Kenya

PONE-D-19-25050R2

Dear Dr. Otiende,

We are pleased to inform you that your manuscript has been judged scientifically suitable for publication and will be formally accepted for publication once it complies with all outstanding technical requirements. Please also make the appropriate minor changes noted by reviewer 2.

With kind regards,

Eric Forgoston

Academic Editor

PLOS ONE

Additional Editor Comments (optional):

Reviewers' comments:

Reviewer's Responses to Questions

**Comments to the Author**

1. If the authors have adequately addressed your comments raised in a previous round of review and you feel that this manuscript is now acceptable for publication, you may indicate that here to bypass the “Comments to the Author” section, enter your conflict of interest statement in the “Confidential to Editor” section, and submit your "Accept" recommendation.

Reviewer #1: All comments have been addressed

Reviewer #2: All comments have been addressed

2. Is the manuscript technically sound, and do the data support the conclusions?

Reviewer #1: Yes

Reviewer #2: Yes

3. Has the statistical analysis been performed appropriately and rigorously? 

Reviewer #1: Yes

Reviewer #2: Yes

4. Have the authors made all data underlying the findings in their manuscript fully available?

Reviewer #1: Yes

Reviewer #2: Yes

5. Is the manuscript presented in an intelligible fashion and written in standard English?

Reviewer #1: Yes

Reviewer #2: Yes

6. Review Comments to the Author

Reviewer #1: (No Response)

Reviewer #2: The authors have sufficiently addressed most of my comments in the revised manuscript.

When defining flat priors on the intercept (line 253), this should read p(\\alpha_d) \\propto 1.

Minor comment: On p12 of the manuscript, prior variances should be defined consistently. e.g. lines 243 - 245 versus line 253.

7. PLOS authors have the option to publish the peer review history of their article (what does this mean?). If published, this will include your full peer review and any attached files.

Reviewer #1: No

Reviewer #2: No

---

## [Editor Report · Acceptance letter]

8 Jun 2020

PONE-D-19-25050R2 

Bayesian hierarchical modeling of joint spatiotemporal risk patterns for Human Immunodeficiency Virus (HIV) and Tuberculosis (TB) in Kenya 

Dear Dr. Otiende:

I'm pleased to inform you that your manuscript has been deemed suitable for publication in PLOS ONE. Congratulations! Your manuscript is now with our production department. 

Kind regards, 

on behalf of

Dr. Eric Forgoston 

Academic Editor

PLOS ONE